# Nanodiamond Effects on Cancer Cell Radiosensitivity: The Interplay between Their Chemical/Physical Characteristics and the Irradiation Energy

**DOI:** 10.3390/ijms242316622

**Published:** 2023-11-22

**Authors:** Veronica Varzi, Emiliano Fratini, Mauro Falconieri, Daniela Giovannini, Alessia Cemmi, Jessica Scifo, Ilaria Di Sarcina, Pietro Aprà, Sofia Sturari, Lorenzo Mino, Giulia Tomagra, Erminia Infusino, Valeria Landoni, Carmela Marino, Mariateresa Mancuso, Federico Picollo, Simonetta Pazzaglia

**Affiliations:** 1Physics Department, National Institute of Nuclear Physics, Section of Turin, University of Turin, Via P. Giuria 1, 10125 Turin, Italy; veronica.varzi@unito.it (V.V.); pietro.apra@unito.it (P.A.); sofia.sturari@unito.it (S.S.); 2Laboratory of Biomedical Technologies, Italian National Agency for New Technologies, Energy and Sustainable Economic Development (ENEA), Casaccia Research Centre, Via Anguillarese 301, 00123 Rome, Italy; emiliano.fratini@enea.it (E.F.); daniela.giovannini@enea.it (D.G.); carmela.marino@enea.it (C.M.); mariateresa.mancuso@enea.it (M.M.); 3Nanomaterials for Industry and Sustainability (NIS) Inter-Departmental Centre, University of Turin, Via Quarello 15/A, 10125 Turin, Italy; lorenzo.mino@unito.it; 4Fusion and Technology for Nuclear Safety and Security Department, Italian National Agency for New Technologies, Energy and Sustainable Economic Development (ENEA), Casaccia Research Centre, Via Anguillarese 301, 00123 Rome, Italy; mauro.falconieri@enea.it; 5Innovative Nuclear Systems Laboratory, Italian National Agency for New Technologies, Energy and Sustainable Economic Development (ENEA), Casaccia Research Centre, Via Anguillarese 301, 00123 Rome, Italy; alessia.cemmi@enea.it (A.C.); jessica.scifo@enea.it (J.S.); ilaria.disarcina@enea.it (I.D.S.); 6Chemistry Department, University of Turin, Via P. Giuria 7, 10125 Turin, Italy; 7Drug Science and Technology Department, University of Turin, Corso Raffaello 30, 10125 Turin, Italy; giulia.tomagra@unito.it; 8Medical Physics Laboratory, IRCCS Istituto Nazionale Tumori Regina Elena, Via Elio Chianesi 53, 00144 Rome, Italy; erminia.infusino@ifo.it (E.I.); valeria.landoni@ifo.it (V.L.)

**Keywords:** nanodiamonds, radiation sensitizer, medulloblastoma, radiation therapy, DNA damage, apoptosis

## Abstract

Nanoparticles are being increasingly studied to enhance radiation effects. Among them, nanodiamonds (NDs) are taken into great consideration due to their low toxicity, inertness, chemical stability, and the possibility of surface functionalization. The objective of this study is to explore the influence of the chemical/physical properties of NDs on cellular radiosensitivity to combined treatments with radiation beams of different energies. DAOY, a human radioresistant medulloblastoma cell line was treated with NDs—differing for surface modifications [hydrogenated (H-NDs) and oxidized (OX-NDs)], size, and concentration—and analysed for (i) ND internalization and intracellular localization, (ii) clonogenic survival after combined treatment with different radiation beam energies and (iii) DNA damage and apoptosis, to explore the nature of ND–radiation biological interactions. Results show that chemical/physical characteristics of NDs are crucial in determining cell toxicity, with hydrogenated NDs (H-NDs) decreasing either cellular viability when administered alone, or cell survival when combined with radiation, depending on ND size and concentration, while OX-NDs do not. Also, irradiation at high energy (γ-rays at 1.25 MeV), in combination with H-NDs, is more efficient in eliciting radiosensitisation when compared to irradiation at lower energy (X-rays at 250 kVp). Finally, the molecular mechanisms of ND radiosensitisation was addressed, demonstrating that cell killing is mediated by the induction of Caspase-3-dependent apoptosis that is independent to DNA damage. Identifying the optimal combination of ND characteristics and radiation energy has the potential to offer a promising therapeutic strategy for tackling radioresistant cancers using H-NDs in conjunction with high-energy radiation.

## 1. Introduction

Radiotherapy (RT) is a medical treatment method that employs ionizing radiation to specifically target and kill cancer cells. In this context, enhancing the effectiveness of RT remains a promising approach. Among the different strategies currently proposed to reduce the detrimental side effects in healthy tissues placed in proximity of the treated volume, the use of radiosensitizers can represent an encouraging line to help in achieving the main optimization of RT treatment [1]. The radiosensitising materials can work selectively, improving the sensitivity to radiation of cancer cells and preventing treatment-related long-term morbidity by reducing the radiation dose and the adverse effects on normal tissues.

To this extent, a new impacting field of research for radiation oncology is nanomedicine, which has become an important development direction of modern medical treatment. Indeed, nanoscale materials provide unique chemical and physical properties that are well-suited for radiosensitising purposes [2,3]. In particular, an innovative, effective and, at the same time, safe approach for improving RT treatment consists of the radiosensitisation of the tumour targets by means of nanoparticles.

In recent decades, the radiosensitising effects of numerous nanoparticle types have been well characterized and there has been a large increase in the number of applications of nanoparticles in medicine, both in the diagnostics and cancer therapy fields. The purpose of them is to induce highly deleterious DNA lesions in cancer cells [4], following the interaction of photons or charged particles with nanoparticles and of subsequent primary ionization and excitation events. Nanoparticles can, thus, intensify the production of secondary electrons and free radicals, like reactive oxygen species (ROS), which are highly reactive chemical species that can cause indirect damage to the cells and in turn enhance RT effects [5]. This approach could allow the delivery of a lower RT dose to the tumour, sparing the surrounding tissue and decreasing long-term side effects especially devastating in paediatric patients.

While the vast majority of studies to date have focused on gold nanoparticles with photon radiation, an increasing number of works are exploring the opportunities offered by other nanoparticles, like gadolinium-, platinum-, iron oxide-, hafnium- [6] and carbon-based ones. Specifically, several in vitro [7] and in vivo [8,9] studies have shown that carbon nanoparticles are non-cytotoxic and well tolerated, and among the possible materials employed to further improve the efficacy of RT, diamond nanocrystals or nanodiamonds (NDs) were highlighted as a new class of carbon nanoparticles really promising in assisting RT in vitro [4]. As proposed, for example, in the study published by Grall et al. [10], a combination of these versatile and promising nanotools and γ photon can increase radiation effectiveness, enhancing the anti-tumour effect of RT in Caki-1 and ZR75.1 (KU70 wt and mut) cells.

Nowadays, NDs can be produced cost-effectively in large quantities, with dimensions ranging from a few nm up to 300 nm, which are suitable for nanoscale applications. NDs are characterized by a high surface-to-volume ratio, which results in the heavy influence of their surface structure on their final properties [11]. Among ND’s outstanding properties, there are a low toxicity, inertness, chemical stability and the possibility of surface functionalization [12,13]. NDs show fluorescence properties thanks to the presence of optically active defects, such as the nitrogen-vacancy centre (NV), consisting of a highly photo-stable fluorescent defect strongly resistant to bleaching and quenching phenomena, which ensures the traceability of these nanoparticles without further functionalization with organic dyes. This appealing property has, therefore, increasingly attracted interest for bioimaging purposes [14,15,16,17]. Furthermore, ND surface moieties can be easily and efficiently modified to achieve the desired specific physical and chemical properties, for example, using high-temperature thermal treatments carried out in a controlled atmosphere [18]. In particular, terminating the ND surface with hydrogen atoms (H-NDs), they exhibit a surface negative electron affinity in aqueous solutions, promoting the transfer of electrons to water molecules. These characteristics ensure a high reactivity with ROSs, increasing the H-NDs’ capability to locally enhance the radiation-induced damage [10]. 

All these features make NDs good candidates for biomedical applications, such as drug delivery [19], tissue engineering [20] and radiosensitizers in RT treatments. With regard to the latter, NDs can emit and produce secondary electrons or radicals upon ‘stimulation’ by ionizing radiation, thus promoting the damaging of the DNA, but it is still unclear whether this damage can be sufficient for explaining the increase in radiation-induced cell death in the presence of NDs [4]. However, even if the studies on ND-mediated radiosensitisation are sparse, there are excellent indications that these nanoparticles represent an interesting subject of examination, due to their potential efficacy in the hydrogenated form (H-NDs) [10]. Hence, further investigations are required to better understand and fully explore the use of NDs in this research field. 

In the present work, the radiosensitising capability of NDs with different chemical/physical characteristics (surface modifications H/OX-NDs, sizes, and concentration) has been tested in vitro on DAOY, a human radioresistant tumour cell line of medulloblastoma (MB). Initially, the ability of MB cells to internalise NDs of different average diameters was assessed using flow cytometry with excitation and emission spectra characteristic of native NV centres present in NDs. Furthermore, to investigate NDs’ cellular uptake and their cellular localization, fluorescence and Raman microscopy were also performed. Next, MB cells incubated with different types of NDs were irradiated with photons of different energies using an X-ray generator operated at 250 kVp or a ^60^Co γ source (1.25 MeV). Increasing doses of photons (2 Gy, 5 Gy and 8 Gy) were delivered in order to build up survival curves through clonogenic cell-survival assay. The objective was to explore how the radiosensitising properties of NDs are influenced by various chemical and physical factors, including size, surface modifications, and concentration, as well as radiation energy. Additionally, we delved into the biological interaction between NDs and radiation, investigating the molecular mechanisms involved, such as DNA damage and apoptosis. The aim was to highlight how chemical/physical features can enhance the RT impact of NDs on brain tumour cells.

## 2. Results

### 2.1. Diffuse Reflectance Infrared Fourier-Transform (DRIFT) Spectroscopy and Photoluminescence (PL) Spectroscopy of ND Samples

In Figure 1A, the DRIFT spectroscopy results for the modified NDs are presented (for details on the treatments, see Table 1 and Appendix A). The spectra of OX-NDs reveal significant signals associated with oxygen-containing functionalities, including bands within the 1850–1650 cm^−1^ spectral region, corresponding to C=O stretching, and a broad band in the 3650–3000 cm^−1^ range, indicative of O–H stretching modes [21,22]. In contrast, the spectra of H-NDs exhibit prominent features in the 2990–2800 cm^−1^ range, associated with C–H stretching [21,22]. Additionally, these spectra lack the O–H stretching signal and show the absence of carbonyl-related fingerprints. 

ND PL spectra are displayed in Figure 1B. They all exhibited broad band spanning the 600–780 nm range, which can be associated with the phonon sideband of NV centres. In some cases, weak features at λ = 575 nm and λ = 638 nm, attributable, respectively, to the zero-phonon line (ZPL) signals of NV^0^ and NV^-^ centres, are also observable [23]. Notably, the intensity of PL in 240 nm NDs exceeded that of 55 nm NDs by approximately one order of magnitude and was greater by about two orders of magnitude than that characterizing 18 nm NDs. It is also worth highlighting that the spectral feature at ~572.5 nm, appearing for all the samples, was not the result of a PL effect; instead, it was the first-order Raman peak of diamond, corresponding to a Raman shift value of 1332 cm^−1^ [24], as can be seen in ND Raman spectra, reported in Appendix A.

### 2.2. Fluorescence Microscopy and Flow Cytometry to Assess ND Internalization

NDs, due to their bright fluorescence and high photostability with sufficiently long lifetime, represent an efficient candidate as markers for cellular imaging and to date, there are a number of publications showing the utilization of ND fluorescence for imaging, mostly in the cells [7,8,14,15,16,25,26].

In the present work, the cellular uptake of hydrogenated and oxidized NDs of different median sizes was investigated (H-NDs and OX-NDs, respectively; Table 1), exploiting the fluorescence properties of the NV colour centres of the NDs. OX-NDs were produced through thermal air oxidation processes performed in a tubular furnace to eliminate the residual graphitic phases and to promote the formation of carboxylic (–COOH) groups on the outer surface of the NDs. H-NDs were obtained via a subsequent thermal hydrogenation process to have covalently bonded hydrogen atoms on the outer surface. Both of these ND variants emitted red fluorescence in the region between 550 nm and 800 nm upon excitation at wavelengths in the green region (around 532 nm) [14,25].

In particular, we employed the fluorescence imaging technique to qualitatively analyse the internalization of NDs in the human MB DAOY cells. We used NDs with a median diameter of 18 nm, 55 nm and 240 nm. Figure 2 shows fluorescence images of DAOY cells, with an overlay of NDs (red), cell nuclei (DNA, blue) and cell cytoskeleton (F-actin, green) emission signals. We reported representative images of the distribution of H-NDs of different sizes and of OX-NDs of 240 nm compared to the ND-untreated control (CTR) (Figure 2A,B).

In line with the observations made by Fu et al. [26], both OX- and H-NDs were primarily localized in the perinuclear region, as shown in Figure 2A,B, as well as in Appendix A. However, a difference emerged when comparing their intracellular distributions. OX-NDs exhibited an even dispersion throughout the cytoplasm, while H-NDs displayed a tendency to cluster, particularly noticeable with larger NDs (240 nm) that were more readily detectable through fluorescence.

Although not experimentally verified, we hypothesized that this differing intracellular localization between H-NDs and OX-NDs could be attributed to the hydrophobic nature of H-NDs, making them more prone to aggregation and adherence to cell membranes. Conversely, OX-NDs, with their hydrophilic surface and carboxylic groups, tended to disperse throughout the cytoplasm. The distinct behaviours of H-NDs and OX-NDs in a biological context may also suggest a higher degree of interaction of H-NDs with the plasma membrane and various organelles.

The internalization of ND variants was confirmed by a series of vertical cross-sectional fluorescence images of ND-treated cells (Figure 2B). Furthermore, we observed that larger NDs displayed a reduced level of aggregation when compared to their smaller counterparts, such as the 18 nm NDs. This observation aligns with the understanding that aggregates typically form due to stronger Van der Waals interactions, which are more pronounced when the nanomaterial size is smaller [27].

Flow cytometry is a well-developed method used for analysis of cellular uptake for fluorescent nanoparticles like quantum dots [28] or for particles labelled by fluorescent dye [29]. Particularly, the usage of fluorescent NDs as cellular markers has been demonstrated using flow cytometry [30,31,32]. Flow cytometry was employed here to evaluate both the influence of ND concentration on their internalization and the dependence of ND uptake on the incubation time. We showed that the intensity of the ND fluorescence signal increased proportionally to their concentration, reaching the maximum at the highest concentration (400 µg/mL; Figure 2C and Appendix A). The internalization process increased with the incubation time up to 9 h post-administration, when the uptake reaches a plateau (Figure 2D). Therefore, 13 h was selected as an appropriate incubation time for the subsequent experiments.

### 2.3. Raman Microscopy to Assess ND Cellular Distribution

In addition to fluorescence imaging, results obtained using Raman microscopy provided valuable information about the distribution of NDs in DAOY cell cultures (Figure 3 and Figure 4). Indeed, Raman mapping can produce direct information about the cellular ND localization. This technique provides the spatial distribution of the Raman signal, constituted by the Raman peaks that belong to specific compounds present in the sample under consideration. In particular, 2D Raman mapping can be used to detect the unique and sharp Raman signal of the sp^3^ diamond structure of NDs at 1332 cm^−1^, and to obtain the positioning of ND probes in cell cultures. Indeed, the diamond Raman peak is an excellent marker since it gives unequivocal evidence of the presence of NDs based on their chemical and structural identification, differently from bright field and fluorescence techniques; therefore, in this work, we used this peak to study the internalization of 240 nm OX- and H-NDs in DAOY cells. To this aim, NDs were incubated at a concentration of 20 μg/mL for 13 h and two representative bright-field images of DAOY cells with OX-NDs and H-NDs were acquired (Figure 3A and Figure 4A).

The 2D Raman maps were obtained by measuring the Raman spectra of the sample on a grid of points with spacing less than 2 μm, carrying out several acquisitions and then considering the average spectrum. In Figure 3A, two exemplary mapped points (I and II) are indicated with a cross and the corresponding Raman spectra are shown in Figure 3B. In point (I), an area without NDs is represented, while in point (II), the presence of NDs can be clearly detected by the Raman peak (highlighted in green). 

After the point-by-point spectra acquisition, we considered the integral of the Raman spectra in the region containing the ND signature, i.e., between 1309 cm^−1^ and 1345 cm^−1^, and obtained the ND peak area by subtracting the luminescence background of the spectra, defined through a baseline between the end-points of the considered Raman shift interval. The results of these operations are shown in Figure 3C and Figure 4B, for the DAOY cells with OX-NDs and H-NDs, respectively. Once the map was obtained, the localization of the NDs was identified from the merged images (bright-field image + Raman map) shown in Figure 3D and Figure 4C. 

Comparing the results obtained for OX- and H-ND maps, stronger diamond signal intensities were detectable for the OX type (Figure 3C and Figure 4B). Moreover, consistently with the fluorescence imaging information, H-NDs formed larger clusters due to their greater tendency to aggregate in water-based solutions with respect to OX-NDs (Figure 4). The ND cellular localization is non-uniform but concentrated in specific regions and Raman mapping confirmed the presence of a large amount of NDs inside the cytoplasm of DAOY cells, both for OX- and H-NDs.

### 2.4. Clonogenic Survival after Combined Treatment with NDs and X-rays

Clonogenic assay is the gold-standard method for measuring the radiosensitivity of cancer cells. To evaluate whether NDs could sensitize tumour cells to radiation-induced cell killing, we tested a combined ND and X-ray (250 kVp) treatment on the cell survival of a DAOY cell line. To this aim, the cells were treated with various concentrations of NDs (40, 20, 10 µg/mL) of different sizes (240, 55, 18 nm) and surface terminations (OX- and H-NDs; Table 1) and then irradiated them with different doses of X-rays (0 Gy, also named sham-irradiated, 2 Gy, 5 Gy and 8 Gy). 

The first result was that H-NDs had a toxic effect on cell viability, as opposed to OX-NDs, which showed a trend toward increasing the survival fraction. In detail, H-NDs of all sizes significantly decreased cell survival at a concentration of 20 µg/mL (240 nm H-NDs *p*-value = 0.004, Figure 5A; 55 nm *p*-value = 0.018, Figure 5B; 18 nm H1 *p*-value = 0.0001 and 18 nm H2 *p*-value = 0.03, Figure 5C), while a minor effect was seen at 10 µg/mL. Moreover, 18 nm H-NDs were the most toxic, probably due to their tendency to aggregate, with a dependence on the different hydrogenation treatment. The higher efficacy of the hydrogenation treatment (6 h, 850 °C) carried out on H2-NDs, resulting in a complete elimination of oxygenated surface moieties, was responsible for their increased hydrophobic behaviour and, indeed, H2-NDs produced a stronger decrease in cell survival compared to H1-NDs (hydrogenation 3 h, 750 °C).

Notably, the combined treatment with H-NDs and X-rays of 250 kVp did not produce a synergic effect (Figure 5A,B) and also OX-NDs did not show a radiosensitising behaviour. Only 18 nm H2-NDs produced a significant decrease in the survival fraction at 2 and 5 Gy (*p*-value ≤ 0.0001, Figure 5C), probably due to the cumulative effect of the radiation treatment to the basal toxicity. The high cell killing induced at 8 Gy prevented the assessment of the efficacy of the combined treatment, and this dose was not replicated in the following experiments.

### 2.5. Clonogenic Survival and DNA Damage after Combined Treatment with NDs and γ-Rays

To test the dependency of the radiosensitising effects of NDs on radiation energy, we replicated the previous experiment, applying 1.25 MeV γ-rays from a ^60^Co source (Calliope Facility, ENEA) and selecting a subset of H-NDs showing lower basal toxicity (240 nm, 55 nm and 18 nm H1 at 10 μg/mL). The clonogenic cell survival confirmed the higher basal toxicity for 18 nm H-NDs seen in the experiments with the X-rays (*p*-value = 0.03, Figure 6A). The combined treatment with 1.25 MeV γ-rays produced a significant decrease in the survival fraction at 5 Gy only for the 240 nm H-NDs, compared to the ND-untreated 5 Gy irradiated control (*p*-value = 0.01, Figure 6A,B). 

In order to assess if the toxic effect of the combined treatment was due to an increase in DNA damage, we evaluated, using flow cytometry, the phosphorylation of histone H2AX (γ-H2AX), a double-strand breaks (DSBs) marker, 30 min and 24 h after irradiation with 2 Gy and 5 Gy. While irradiation with 2 Gy did not produce any increase in γ-H2AX over the unirradiated cell level at 30 min post-irradiation, we report a general significant γ-H2AX increase after irradiation with 5 Gy (*p*-values: CTR-0 Gy vs. all 5 Gy samples ***; 240 H-0 Gy vs. all 5 Gy samples **; 55 H-0 Gy vs. all 5 Gy samples ***; 18 H-0 Gy vs. all 5 Gy samples **; Figure 6C). Significant differences were also observed between 2 Gy and 5 Gy irradiated cells for nearly all the treatments (*p*-values: CTR-2 Gy vs. all 5 Gy samples *; 55 H-2 Gy vs. all 5 Gy samples **; 18 H-2 Gy vs. all 5 Gy samples **; Figure 6C), with the exception of the 240 nm H-ND samples. No differences in γ-H2AX were observed within each dose group (unirradiated samples, and 2 Gy and 5 Gy irradiated samples), indicating that treatment with H-ND does not affect the induction of DSBs. Furthermore, the expression of γ-H2AX at 24 h post-irradiation was restored to the basal level for all treatments, excluding the possibility of a higher complexity of DNA damage for H-NDs (Figure 6D).

### 2.6. Bax and Caspase-3 Analysis after 240 nm H-ND/γ-Ray Combined Treatment

Since no significant increase in DNA damage has been detected for the 240 nm H-ND/γ-ray combined treatment, we further assessed the protein content for pro-apoptotic markers Bax and cleaved-Caspase-3 at 3 h post-irradiation (Figure 7). Although all 240 nm H-ND combined treatments showed a trend toward a dose-dependent increase in cleaved-Caspase-3 expression, only the 5 Gy γ-ray combined treatment reached significance compared to both the untreated controls and 240 nm OX-ND-treated samples (*p*-value = 0.04 for both of them; Figure 7B). This is in accordance with our clonogenic survival results. Instead, Bax was unmodulated in all the treatment conditions at both the protein (Figure 7C) and transcript expression levels (Appendix A). These results indicated the involvement of a Bax-independent Caspase-3 activation in increased cell killing for the 240 nm H-ND/γ-ray combined treatment.

## 3. Discussion

RT plays a critical role in the management of many tumour types. In MB, RT is typically performed after surgery, in combination with chemotherapy. C = The combination of different therapeutic strategies may enhance the efficacy and reduce the toxicity observed in high-dose RT. As mentioned, it has been proven that the use of radiosensitising nanoscale materials can improve the radiation sensitivity of cancer cells and reduce adverse effects. Here, we tested the potential efficacy of NDs of different sizes and surface modifications, in combination with irradiation at different energies, in radiosensitising a human radioresistant MB cell line (DAOY).

Concerning the characterizations of the modified NDs, the results of the FTIR spectroscopy analysis not only confirm the success of the oxidation processes, but also underscore the high hydrophilicity of OX-NDs [11,33,34], as the O–H stretching band is also influenced by the presence of surface-adsorbed water [21,22]. Indeed, the hydrophilicity of OX-NDs was also highlighted by DLS measurements (Appendix A), displaying a high dispersibility of these particles in aqueous solutions. The DRIFT spectra for H-NDs also show the efficacy of hydrogenation treatments in forming surface hydrogen terminations, while simultaneously removing oxygenated functional groups, thus turning the surface from hydrophilic to hydrophobic [11,33]. Moreover, ND fluorescence assessed through PL spectroscopy revealed that the intensity of the ND PL spectra was directly correlated to the sizes of the NDs. Specifically, the 240 nm NDs characterized by a larger diamond core, with a higher concentration of NV centres, generated a stronger PL signal compared to the 55 nm and 18 nm NDs.

After ND characterization, we investigated the cellular uptake and distribution of NDs in DAOY cells. The uptake kinetic showed good internalization as well as retention in MB cells, in line with the kinetic properties of other nanoparticles [35,36,37], with a saturation point at 9 h post administration. The internalization of NDs was confirmed by both fluorescence and Raman microscopy. Both H- and OX-NDs localized mainly in the perinuclear area, with some difference in smaller H-NDs that tended to form aggregates and stick on cell membranes due to their hydrophobic nature.

After confirming the uptake of NDs in DAOY cells, our next step was to examine the effect of NDs as radiosensitizers and the dependency of this effect on their characteristics such as surface modification, dimensions, and concentration. The first finding was that H-NDs alone reduced the viability of DAOY, revealing a basal cytotoxicity of H-NDs that was inversely proportional to their size. In particular, the higher toxicity of the smallest NDs (18 nm) might be explained both by their greater tendency to aggregate and stick on cellular membranes and by the higher amounts of graphitic phases on their surface, which decrease their biocompatibility [34,38]. 

Cytotoxicity was also influenced by the concentration of NDs, and in line with reduced ND aggregation, less toxicity was observed at the lowest H-ND concentration (10 µg/mL). Nevertheless, our results did not reveal a straightforward correlation between ND concentration and cytotoxicity, which aligns with several studies that have reported conflicting outcomes [39,40,41,42].

As regards the administration of combined treatments with NDs and irradiation, we highlighted an opposite behaviour of OX- and H-NDs, with the latter enhancing radiation-induced cell killing. Two main hypotheses could be proposed: either the electrons are emitted by the NDs and are sufficiently energetic to generate secondary radiolysis, or this overproduction is based on interfacial processes. In the former case, the difference between H-NDs and OX-NDs can be ascribed to their different surface chemistries (hydrogen terminations for H-NDs versus oxidized terminations for OX-NDs). Following excitation by radiation, the negative electron affinity of H-NDs could promote the transfer doping of electrons from the valence band to the redox species in the cytoplasm, and the high reactivity of these emitted electrons with radical species can, in turn, increase the indirect radiation damage. The second hypothesis is linked to the key role that water molecules at the nanoparticle interface could play in the production of radicals. Indeed, for NDs, considering the mass energy absorption coefficients of carbon and water, we do not expect a drastic increase in the energy deposition in correspondence with these NDs, both for keV and MeV irradiation. However, it has been documented that water molecules adsorb at a lower vapour pressure on H-NDs with respect to OX-NDs, leading to a higher amount of water on H-NDs despite their hydrophobicity [34,43]. Furthermore, by combining infrared, Raman and X-ray absorption spectroscopies, Petit et al. revealed different structures of water molecules surrounding the NDs when the surface chemistry is varied [44]. In particular, the hydrogen bonding network of water was found to be different in aqueous dispersions of H-NDs compared to OX-NDs, leading to long-range disorder of the water molecules and electron transfer in the shell of H-ND hydration. Thus, a difference in water disposition could also be responsible for the different behaviours of H- and OX-NDs under irradiation [45].

Focusing on the effect of H-NDs, we found differences in cell survival dependent on the intensity of the employed beam energy. While at low energies (250 kVp) a synergic effect of the combined treatment was generally not seen, with 1.25 MeV γ-rays we found a significant decrease in cell survival at the 5 Gy dose co-treating with 240 nm H-NDs.

Most studies in recent decades have widely focused on the radiosensitising effects of metallic nanoparticles, in particular gold ones, combined with photon irradiation. In this case, the high atomic number (Z) of the metallic nanoparticles gives them a significantly higher X-ray mass energy-absorption coefficient compared to biological tissues [46]. These effects were maximized after exposure to low linear energy transfer (LET) radiation in the range of kilovoltage photon energy [47], where the predominant physical interaction between photons and nanoparticles is the photoelectric effect. In our study, where 240 nm H-NDs were able to radiosensitise cells in combination with 5 Gy of γ-rays at 1.25 MeV, the Compton effect was predominant in driving the photon radiation–matter interaction. Previously, Grall et al. proposed [10] H-NDs to increase the efficacy of radiation on different cancer cell lines, supporting the idea of using nanoparticles not necessarily composed of high-Z atoms to improve RT treatments. 

Based on their physical characteristics, H-NDs present several advantageous properties for radioenhancement, as they benefit from their semiconductor behaviour and negative electron affinity with a positive charge in water solutions [10], which favour H-NDs in the direct photo-induced emission of low-energy secondary electrons from their surface into the surrounding medium (electrons can escape once excited in the conduction band). These properties ensure high H-ND reactivity with oxygen species in an aqueous environment as the cellular one, and allow them to emit secondary electrons, thus acting as a potential source of ROS when activated by the photoemission of electrons via ionizing radiation. 

The last step was the investigation into the nature of H-ND–radiation biological interactions and the molecular mechanisms underlying the H-ND-induced radiosensitisation. We showed that pre-treatment of DAOY cells with 240 nm H-NDs in combination with 5 Gy γ-ray irradiation (1.25 MeV) significantly increased cell killing and this was associated with a Bax-independent increase in Caspase-3 cleavage at 3 h post-irradiation. However, DNA damage was not involved in radiosensitisation, since radiation-induced γ-H2AX phosphorylation was not significantly increased by treatment with H-NDs of any size. In addition, the DNA damages were completely restored after 24 h, confirming the absence of slowly repairable clustered damages.

Many studies employing different nanoparticles as radiosensitizers suggest that nuclear DNA is the main target of the nanoparticle-mediated radiosensitisation of ionizing radiation and that nanoparticles effects rely on the increase in DNA damage, correlating the increase in cell killing with higher induction of DNA DSBs in irradiated cells pre-incubated with different types of nanoparticles [48,49,50]. In particular, the in vitro studies by Grall et al. reported increased DSBs and cell senescence in several radioresistant cancer cell lines treated with H-NDs and γ-radiation (^137^Cs, 4 Gy) [10].

On the other hand, consistently with our results, many reports have failed to demonstrate an increase in nanoparticle-mediated DSB damage in irradiated cells, although a significant radiosensitising effect had occurred [51]. The discrepancy also exists for nanoparticles with very similar chemical/physical parameters [51,52]. The molecular mechanism of the radiosensitising effect of nanoparticles is still unknown and represents a subject of intensive controversy due to the fact that the results on the involvement of nuclear DNA in this mechanism are contradictory.

In our experimental setup, the application of 240 nm H-NDs in conjunction with a 5 Gy dose of γ-rays at 1.25 MeV to the radioresistant DAOY cell line resulted in a substantial increase in Caspase-3 cleavage, which closely correlated with the reduction in cell viability seen following the same treatment. From a mechanistic perspective, our data provided solid evidence supporting the role of terminal apoptosis in the process of radiosensitisation by NDs. Notably, terminal apoptosis was unrelated to the induction of DNA damage, as indicated by the absence of any significant modulation of γ-H2AX and Bax. It is worth mentioning that while Bax is not a direct marker for DNA damage, its presence suggests an apoptotic response triggered by DNA damage.

Even though nuclear DNA is considered the main target for ionizing radiation [51,53], our findings show that NDs localized in the cytoplasm and did not penetrate the cell nucleus. Therefore, although the possibility that the interaction between nanoparticles and radiation might produce secondary electrons/radicals occasionally reaching the nucleus and damaging DNA may not be completely ruled out, it is unlikely that this damage can account for the increase in radiation-induced cell death by nanoparticles. Indeed, the reactive radicals and secondary electrons produced by irradiated nanoparticles are short-lived and can only act in a limited spatial range. Therefore, these damaging agents can only concentrate at high levels in tight shells around nanoparticle clusters. Thus, as suggested by an activation of the Bax-independent Caspase-3 signalling pathway, alternative targets for nanoparticles-mediated radiosensitisation may exist, and these should be sought based on the localization of intracellular nanoparticle hotspots. 

Apoptosis can be activated by both exogenous stimuli and signalling (extrinsic pathway) and endogenous unrepairable damage, clustered DNA damage, organelles and, in particular, mitochondrial damage that is central in the intrinsic pathway (also called mitochondrial apoptosis). DAOY radioresistance, similarly to many radioresistant tumours, is associated with a P53-dependent impairment in the intrinsic pathway; thus, H-ND mediated apoptosis should be activated downstream of the p53-Bax axis [54,55,56].

Direct mitochondrial outer-membrane permeabilization is one of the main effectors in radiation-induced cell death in tumour cells [57], but other molecular targets in the intrinsic or extrinsic pathway are likely to be involved in the radiosensitising action of H-NDs in RT (Figure 8).

In summary, by applying fluorescence and Raman microscopy and flow cytometry, we showed the good internalization and cytoplasm localization of all ND variants. In addition, our findings highlighted the complexity of the radiosensitising effect of NDs in RT. We show that ND features, including chemo-physical modification (H or OX), size and the concentrations, strongly influence the cell viability and survival of irradiated cells. In addition, we demonstrated that radiation energy is also a key factor for the radiosensitising effects of NDs. Moreover, the molecular mechanisms of ND radiosensitisation, involving the activation of Bax-independent Caspase-3, but not DNA damage, remain to be fully characterized, as well as the nature of molecular targets involved. Further research is necessary to unravel the complexity of ND effects and clarify the molecular mechanisms of ND biological interactions before clinical application of NDs in RT.

## 4. Materials and Methods

### 4.1. Cell Cultures

Human MB cell lines (DAOY HTB-186) were obtained from American Type Culture Collection (ATCC; Manassas, VA, USA). Cell lines were routinely maintained in the complete growth medium Eagle’s Minimum Essential Medium (MEM) with 2 mM glutamine and 100 U penicillin/0.1 mg/mL streptomycin, supplemented with 10% foetal bovine serum. Cells were cultured in standard CO_2_ incubators in an atmosphere of 2% oxygen, 5% carbon dioxide, and balanced nitrogen.

### 4.2. ND Preparation

Micron + NDs with a median diameter of 240 nm were purchased from the manufacturer ElementSix™ (Harwell Oxford, UK), while MSY NDs with a median size of 55 nm and 18 nm were manufactured by Pureon (Lengwil, Switzerland). All NDs were obtained by milling high-pressure high-temperature (HPHT)-type Ib single crystals. The ND surface was modified and treated by thermal treatments in a tube furnace [34]. All samples were previously subjected to an annealing treatment carried out at temperatures around 800 °C in an inert atmosphere of N_2_ flux for 2 h. This thermal process aimed to eliminate the heterogeneous surface functionalities of the untreated NDs to standardize their surface. At the same time, it was employed to graphitize amorphous carbon components on the surface without damaging the diamond phase. NDs were thus made suitable for the following processing steps, while also forming new NV centres, thanks to the high-temperature-induced pairing of nitrogen impurities and native vacancies. Following the annealing treatment, an oxidation (etching) process was carried out in air at temperatures between 475 °C and 500 °C for times ranging from 12 to 18 h to obtain the oxidized NDs (OX-NDs). Hydrogenation of the H-NDs was carried out in hydrogen flux at temperatures between 750 °C and 850 °C for 3 or 6 h on NDs that had previously undergone annealing and oxidation.

Table 1 reports the labels used to name univocally each sample and summarizes all the different thermal processing steps carried out on a specific batch of NDs.

### 4.3. DRIFT Spectroscopy

To assess the efficacy of oxidation and hydrogenation thermal treatments performed on NDs, DRIFT spectroscopy was carried out. Spectra were collected under dry air conditions, by using a Bruker Vector 22 FTIR spectrometer equipped with a mercury–cadmium–telluride detector. Each spectrum was registered by averaging 64 interferograms at a spectral resolution of 2 cm^−1^. The recorded reflectance data were subsequently transformed into pseudo-absorbance values: A = −log(R), where R represents the measured reflectance.

### 4.4. PL Spectroscopy

PL spectroscopy was employed to investigate the optical properties of NDs, by assessing the fluorescence arising from NV centres. For PL data collection, the NDs were dispersed in isopropanol and deposited onto a silicon wafer substrate. PL spectra were acquired using a Horiba Jobin Yvon HR800 microspectrometer equipped with a continuous Nd-YAG 532 nm excitation laser and a CCD detector, incorporating a Peltier cooling system (−70 °C). Thanks to the insertion of a filter along the optical path, it was possible to set the power intensity of the laser to 1.69 mW, which was focused by means of a 20× objective, enabling us to probe a sample area of 10 × 10 μm^2^ at a confocal depth of ~3 μm. The employed 600 lines mm^−1^ diffraction grating allowed us to achieve a 3 cm^−1^ spectral resolution.

### 4.5. ND Administration to the Cell Cultures

Prior to incubation in DAOY cells, the NDs were resuspended in bidistilled water at 1.6 mg/mL, autoclaved at 121 °C for 15 min and sonicated for 30 min. Sonicated NDs were added into the cell culture dish, where the cells were previously plated to have the desired final concentration (400, 200, 100, 50, 40, 20, 10, 0 µg/mL).

### 4.6. Visualization of ND Location Using Fluorescence Microscopy

MB cells were plated on 20 mm glass bottom cell culture dish (NEST) and treated for 13 h with NDs in MEM with 10% FBS at 37 °C. ND-treated cells were fixed with 4% Paraformaldehyde *w*/*o* methanol in phosphate-buffered saline (PBS) for 10 min, permeabilized with 0.1% Triton X-100 in PBS for 3 min, blocked with 1% Bovine serum albumin (BSA) for 30 min and stained with 1.65 µM Alexa Fluor™ 488 Phalloidin (Invitrogen, 518 nm—green emission) to selectively label F-actin, 1 µg/mL 4′,6-diamidin-2-fenilindolo (DAPI, 470 nm—blue emission) to visualize nuclei labelling DNA, and 1% BSA in PBS for 20 min in the dark. At last, cells were washed twice with PBS and mounted with Vectashield mounting medium for fluorescence (Vector Laboratories, Newark, CA, USA). The cells were imaged and analysed with an Axio Observer 7 inverted microscope (ZEISS, Oberkochen, Germany), using the following filter sets: 96 HE, 489096-9100-000 (excitation Bandpass BP 390/40, beamsplitter FT 420, emission BP 450/40) for DAPI emission detection; 38, 1031-346 (excitation BP 470/40, beamsplitter FT 495, emission BP 525/50) for Alexa Fluor 488 emission detection; 00, 488000-0000-000 (excitation BP 530-585, beamsplitter FT 600, emission LP 615) for ND emission detection.

### 4.7. Visualization of ND Location Using Raman Microscopy

Raman microscopy was performed to detect the presence of NDs in the biological media and to obtain information on their localization with respect to the cultured cells, through the sharp and intense Raman signature of NDs at 1332 cm^−1^. DAOY cells were plated on a 20 mm glass-bottom cell culture dish (NEST) and treated for 13 h with NDs. ND-treated cells were fixed with 4% Paraformaldehyde *w*/*o* methanol in PBS for 10 min, washed with PBS and left in PBS solution. The Raman setup is a homemade confocal system using 532 nm laser excitation. The laser beam is routed to the cells fixed in the cell culture dish via a metallographic microscope equipped with a 100× immersion objective and with a sample holder with computer-controlled positioning. The backscattered radiation is collected by the same objective, then the elastic component is removed by an edge filter and the surviving signal is fed to a 550 mm monochromator (Horiba-Jobin-Yvon TRIAX 550) matched with a liquid-nitrogen cooled CCD detector (Horiba-Jobin-Yvon CCD-3000) for dispersion and detection. Raman maps were collected using 10 s acquisition time and 3 averaged acquisitions per point with a position step of approximately 2 μm.

### 4.8. Cell Irradiation with X-rays

Cells were irradiated with a Gilardoni CHF 320G X-ray generator operated at 250 kVp, 15 mA, delivering doses of photons of 2, 5 and 8 Gy, with a dose rate of 0.8 Gy/min. Six-well plates or 25 cm^2^ flasks were irradiated in a horizontal position (radiation incident perpendicularly from above). The DAOY cells were returned to the incubator at 37 °C immediately after exposure to radiation, before further processing.

### 4.9. Cell Irradiation with γ-Photons

The γ-irradiations were carried out at the Calliope irradiation Facility [58], located in the ENEA Casaccia Research Centre in Rome. It consists of a pool-type irradiation plant equipped with a ^60^Co (mean energy 1.25 MeV) radio-isotopic source array in a high volume (7.0 m × 6.0 m × 3.9 m) shielded cell. The source rack is planar in shape, with 25 ^60^Co source rods (active area: 41 cm × 75 cm). The MB cells were subjected to γ-radiation at a dose rate of 0.4 Gy/min with final absorbed doses of 2 and 5 Gy. The well plates were irradiated at a horizontal position. Even in this case, cells were returned to the incubator at 37 °C after exposure to γ-radiation and before further processing.

### 4.10. Clonogenic Survival

The day prior to initiating the cell treatment, we conducted a count of viable cells using the trypan blue exclusion test. Subsequently, the cells were distributed into 6-well cell culture plates. Specifically, we seeded 100 viable cells per well for the sham group and the 2 Gy irradiation group, 400 cells per well for the 5 Gy group, and 2000 cells per well for the 8 Gy group. Thirteen hours prior to irradiation, we introduced NDs into the culture medium. Just before subjecting the cells to irradiation, we replaced the medium with a fresh one devoid of NDs. Eight days after treatment, colonies were washed with PBS and fixed and stained with 1% crystal violet (Sigma-Aldrich, Wicklow, Ireland) solution/methanol (1:1). Colonies with >50 cells were counted; the ratio of the number of colonies to the number of cells seeded determined the plating efficiency (PE). The number of colonies that arise after treatment of cells, expressed in terms of PE, is called the surviving fraction.

### 4.11. Flow Cytometry Analysis

ND uptake analysis: MB cells, treated with different concentration of NDs (400, 200, 100, 50, 20, 0 µg/mL) were washed with PBS, harvested and fixed with 4% Paraformaldehyde in PBS for 10 min. Samples were analysed using a CytoFLEX flow cytometer (Beckman Coulter, Brea, CA, USA). The single-cell population was gated in a plot of FSC versus SSC after excluding cell debris and doublets and detected with a yellow-green excitation laser (561 nm) and a 712/20 nm bandpass filter. Data were analysed using FCS Express 7 (De Novo Software, Pasadena, CA, USA) and represented as the Median Fluorescence Intensity (MFI) normalized on the untreated controls.

γ-H2AX assay: MB cells were treated with combined treatment (NDs + radiation) as described above. Cells were washed with PBS, harvested and fixed with 70% ethanol solution at 30 min and 24 h after irradiation. The cells were then washed in Tris-buffered saline (TBS), pH 7.4, and suspended in 0.1% Triton X-100 in a 4% solution of Foetal bovine serum (FBS) in TBS (TST) to suppress nonspecific Ab binding. The cells were then incubated for 2 h at room temperature with Phospho-Histone H2A.X (ser139) Rabbit Ab (Cell Signaling Technology, MA, USA) diluted 1:500 in TST and 45 min with F(ab’)2-Goat anti-Rabbit IgG (H + L) Cross-Adsorbed Secondary Antibody, Alexa Fluor™ 488 (Invitrogen, Waltham, MA, USA), diluted 1:2000 in TST. Washes after each step were performed with a 2% solution of FBS in TBS. At last, cells were suspended in 1 µg/mL of DAPI + 100 µg/mL DNase-free RNase A in TBS. Samples were analysed using a CytoFLEX flow cytometer (Beckman Coulter, Brea, CA, USA) with violet (405 nm) and blue (488 nm) excitation lasers and 450/45 (DAPI) and 525/40 (γ-H2AX) bandpass filters, respectively. Data were analysed using FCS Express 7 (De Novo Software). Cell population was gated in a plot of FSC versus SSC to exclude cell debris and subpopulations were gated in a plot of SSC versus 450/45 fluorescence intensity to identify cells in the G0/G1-, S- and G2/M- phases. For each cell cycle phase, the MFI at 525/40 was calculated separately and normalized by presenting per unit of DNA by dividing the MFI of S- and G2/M-phase cells by 1.5 and 2.0, respectively. Data were represented as the MFI per unit of DNA normalized on the sham-irradiated untreated controls.

### 4.12. Western Blot

DAOY cells were seeded in BioLite 25 cm^2^ flasks (Thermo Scientific™, Wilmington, DE, USA) containing complete growth medium MEM and incubated 13 h with NDs. Before irradiation, the medium was discarded and the cells were washed with PBS, followed by the replacement of fresh medium. At 3 h post-irradiation, cells were rinsed with PBS and collected after trypsin digestion and centrifugation. The collected cells were then treated with T-PER Tissue Protein Extraction Reagent (Pierce Biotechnology, Waltham, MA, USA) with protease inhibitors for protein collection and further quantification through Bio-Rad Protein Assay (Bio-Rad^®^ Laboratories, Hercules, CA, USA). Proteins were separated on 10% pre-cast gels (Mini-PROTEAN TGX Gels, Bio-Rad^®^ Laboratories), transferred onto a 0.2 μm polyvinylidene fluoride (PVDF) membrane (Trans-Blot Turbo Transfer Pack, Bio-Rad^®^ Laboratories,) and blocked with 5% bovine serum for 1 h in Tris-buffered saline with 0.05% Tween^®^-20 (TBST). Membranes were then incubated at 4 °C overnight with Anti-cleaved-Caspase 3 (ASP175) (5A1E) rabbit polyclonal antibody (Cell Signalling Technology^®^, Danvers, MA, USA) and Anti-Bax (ab7977) rabbit polyclonal antibody (Abcam, Cambridge, UK) in 3% bovine serum in TBST and for 1 h at Room Temperature with Anti-β-actin (#A5441) mouse monoclonal antibody (Sigma Aldrich, St. Louis, MO, USA) in 3% bovine serum in TBST. Then, samples were further incubated with horseradish peroxidase (HRP)-conjugated anti-rabbit IgG and anti-mouse IgG for 1 h at room temperature. Specific proteins were visualized using ECL™ Prime Western Blotting Detection Reagent (RPN2232, Cytiva Marlborough, MA, USA) with iBright Imaging Systems and iBright Analysis software version 5.2.0.

### 4.13. Real-Time PCR

DAOY cells were seeded in BioLite 25 cm^2^ flasks containing complete growth medium MEM and incubated 13 h with NDs. Before irradiation, the medium was discarded and the cells were washed with PBS, followed by the replacement of fresh medium. At 3 h post-irradiation, cells were rinsed with PBS and collected after trypsin digestion and centrifugation. Total RNA was isolated from each sample using the mini RNeasy kit (QiaGen GmbH, Hilden, Germany). RNA concentration and purity were determined by measuring absorbance using a NanoDrop 2000 Spectrophotometer (Thermo Fisher Scientific, Wilmington, DE, USA). A weight of 1 μg of total RNA was run on a 1% denaturing gel to verify RNA integrity. A weight of 1 µg of total RNA was reverse-transcribed using a IScriptTM cDNA Synthesis Kit (BioRad, Hercules, CA, USA). Real-time PCR was carried out using a BioRad CFX96 TouchTM Real-Time PCR Detection System using SsoAdvanced Universal SYBR Green super Mix (BioRad) and specific primers (Table 2). The expression level of each mRNA was assessed using the ∆∆CT method, and Gadph was used as the housekeeping gene for normalization.

### 4.14. Statistics

Statistical analysis was conducted with two-Way ANOVA with the Greenhouse–Geisser correction and Tukey’s multiple comparison test using GraphPad Prism 8.0.2 (Software, San Diego, CA, USA). Values are represented as mean ± SEM. *p*-values ≤ 0.05 (*) and ≤0.01 (**) were considered as statistically significant; *p*-values ≤ 0.001 (***) and ≤0.0001 (****) were considered as highly statistically significant.

## 5. Conclusions

NDs are considered a new class of carbon nanoparticles promising in assisting RT. Although the toxicity of NDs differs for each nanoparticle type, generally NDs have been shown to be non-toxic or only mildly toxic. Nonetheless, we showed a considerable influence of NDs’ chemical and physical attributes, including their size and concentration, on their toxicity and radiosensitising capacity. While this poses a challenge for research into their effects, it also presents an opportunity, as manipulating the ND chemical/physical surface properties holds the potential to purposefully modify their behaviour in biological contexts. Our research has shown that H-NDs are the most potent radiosensitizers, but there is still potential for optimizing their efficacy in combination with radiation therapy through further refinement of surface modification protocols, such as increasing hydrogenation.

Additionally, we have documented a correlation between the H-ND radiosensitising effects and increased radiation energy. While metallic nanoparticles demonstrated their highest radiosensitisation properties when subjected to low linear energy transfer (LET) radiation within the kilovoltage-photon-energy range, H-NDs exhibit a radiosensitisation effect when exposed to high-energy mega-voltage beams. Notably, beams ranging from 1 to 25 MV are by far the most commonly used for the treatment of deep-seated tumours (>2 cm deep), such as brain tumours, making H-NDs an interesting opportunity for combined treatment.

Finally, while our research has established that the radiosensitising impact of H-NDs operates through a Caspase-3-dependent mechanism that is independent from DNA damage, there is a requirement for more comprehensive investigations to fully grasp the molecular mechanisms and specific targets of the biological interactions involving NDs. On the whole, our investigations suggest that potential lies in discovering the ideal synergy between ND characteristics and radiation energy, which could present a promising therapeutic approach for addressing radioresistant cancers through the utilization of high-energy radiation alongside H-NDs.

## Figures and Tables

**Figure 1 ijms-24-16622-f001:**
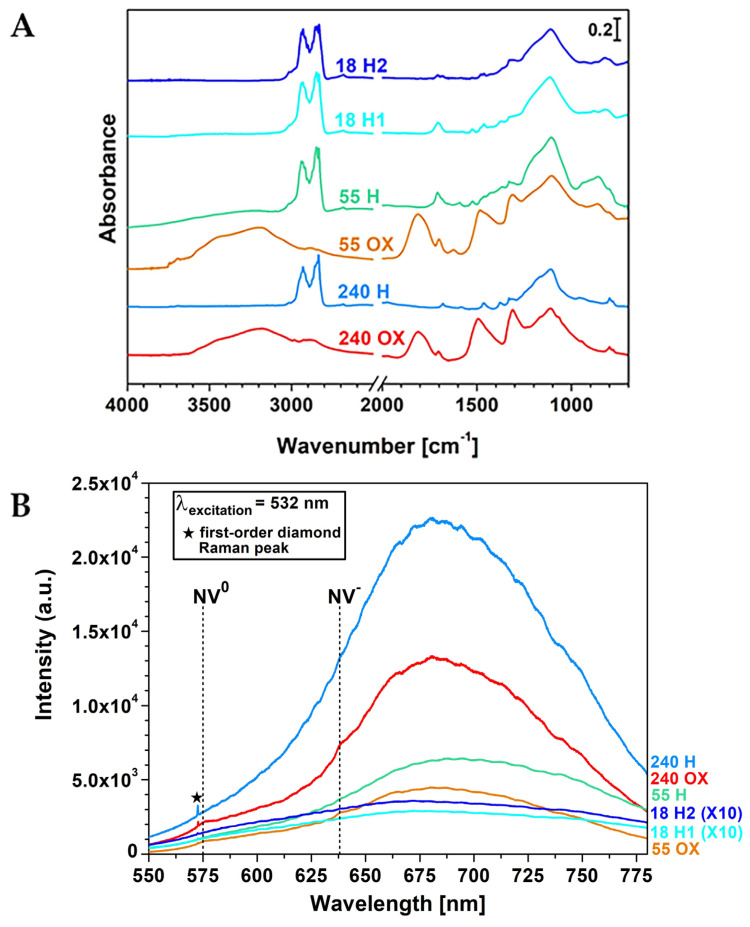
(**A**) DRIFT spectra of the OX- and H-NDs. (**B**) OX- and H-ND PL spectra. The spectra labelled as “18 H2” and “18 H1”, which are referred to the 18 nm NDs (see Table 1), were multiplied by 10 to render them visible on the same scale as the other ones. NV^0^ and NV^−^ zero-phonon lines are marked with black dashed lines. The Raman first-order peak of diamond, appearing in PL spectra at ~572.5 nm, is instead evidenced by a black star.

**Figure 2 ijms-24-16622-f002:**
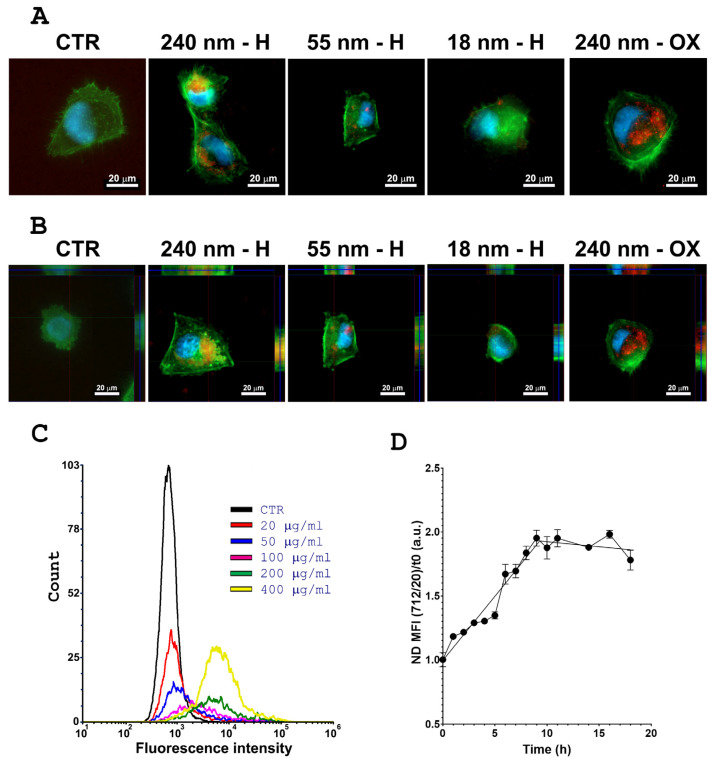
Qualitative analysis of the cellular uptake of NDs by human DAOY cells through fluorescence microscopy: (**A**) Comparison of DAOY cells after 13 h incubation in MEM containing FBS (10% unless otherwise specified) at 37 °C, respectively, without NDs (CTR), 240 nm H- and OX-NDs, 55 nm H-NDs and 18 nm H-NDs, at a particle concentration of 20 μg mL^−1^. The images show the natural fluorescence signal of NDs in red overlapped with the stained nuclei in blue (DAPI) and cell cytoplasm in green (Phalloidin). The scale bar is 20 μm. (**B**) Cross-sectional fluorescence images of the treated and untreated (CTR) DAOY cells. (**C**) Flow cytometry Analysis for MB cells treated with different concentrations of 55 nm H-NDs (CTR black, 20 µg/mL red, 50 µg/mL blue, 100 µg/mL purple, 200 µg/mL green, and 400 µg/mL yellow). Relative fluorescence in the 712/20 channel is reported in the *x*-axis while the number of events is reported in the *y*-axis. (**D**) Uptake curve for 20 µg/mL 240 nm H-NDs analysed using flow cytometry and represented as the Median Fluorescence Intensity (MFI) normalized to the untreated controls.

**Figure 3 ijms-24-16622-f003:**
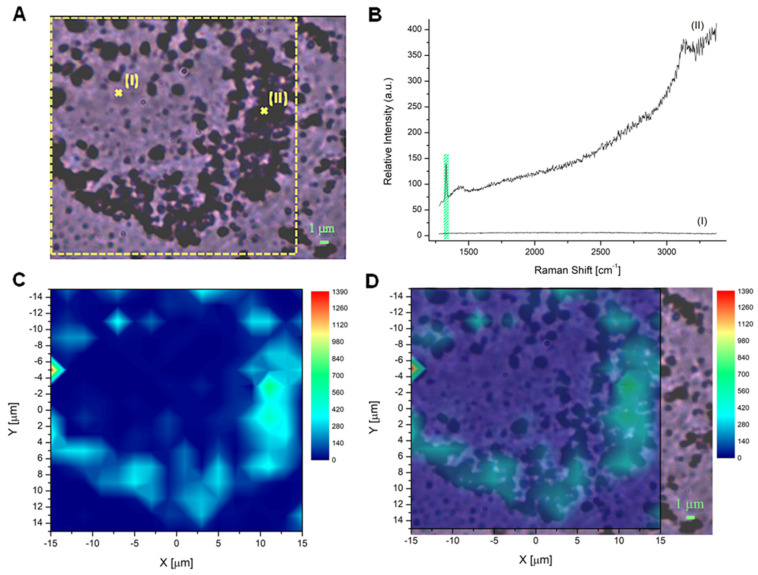
OX-ND localization in DAOY cell cultures using Raman mapping. (**A**) Bright-field image of a cell incubated with a concentration of 20 μg mL^−1^ 240 nm OX-NDs. The yellow dotted square represents the area considered for mapping (scale bar 1 μm). (**B**) Raman spectra measured in point (I) and point (II) in (**A**), showing the typical first-order diamond Raman peak around 1332 cm^−1^ in point (II). The area delimited by the green rectangle represents the Raman shift interval between 1312 cm^−1^ and 1345 cm^−1^, considered to define the baseline to be subtracted from the data in order to avoid the luminescence background. The same interval was employed to evaluate the peak integral and for the subsequent reconstruction of the map reported in (**C**). (**C**) The 16 × 16 Raman map (resolution < 2 μm) of the intensity distribution of the diamond Raman peak, evaluated as the area of the peak in the region of the Raman shift between 1312 cm^−1^ and 1345 cm^−1^. The area covered by the map corresponds to the yellow square outlined in (**A**). (**D**) Merged image of (**A**,**C**).

**Figure 4 ijms-24-16622-f004:**
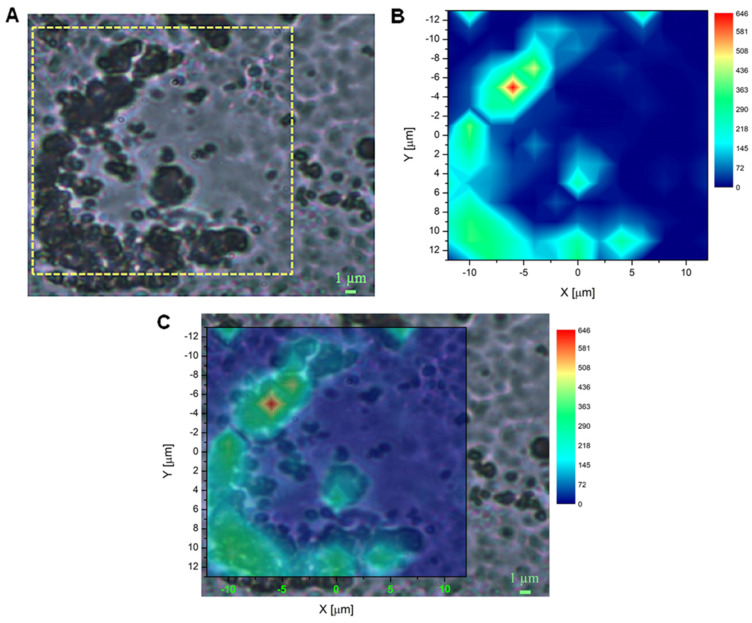
H-ND localization in DAOY cell cultures using Raman mapping. (**A**) Bright-field image of a cell incubated with a concentration of 20 μg mL^−1^ 240 nm H-NDs. The yellow dotted square represents the area considered for mapping (scale bar 1 μm). (**B**) The 13 × 14 Raman map (resolution < 2 μm) of the intensity distribution of the diamond Raman peak, evaluated as the area of the peak in the region of the Raman shift between 1309 cm^−1^ and 1335 cm^−1^. The area covered by the map corresponds to the yellow square outlined in (**A**). (**C**) Merged image of (**A**,**B**).

**Figure 5 ijms-24-16622-f005:**
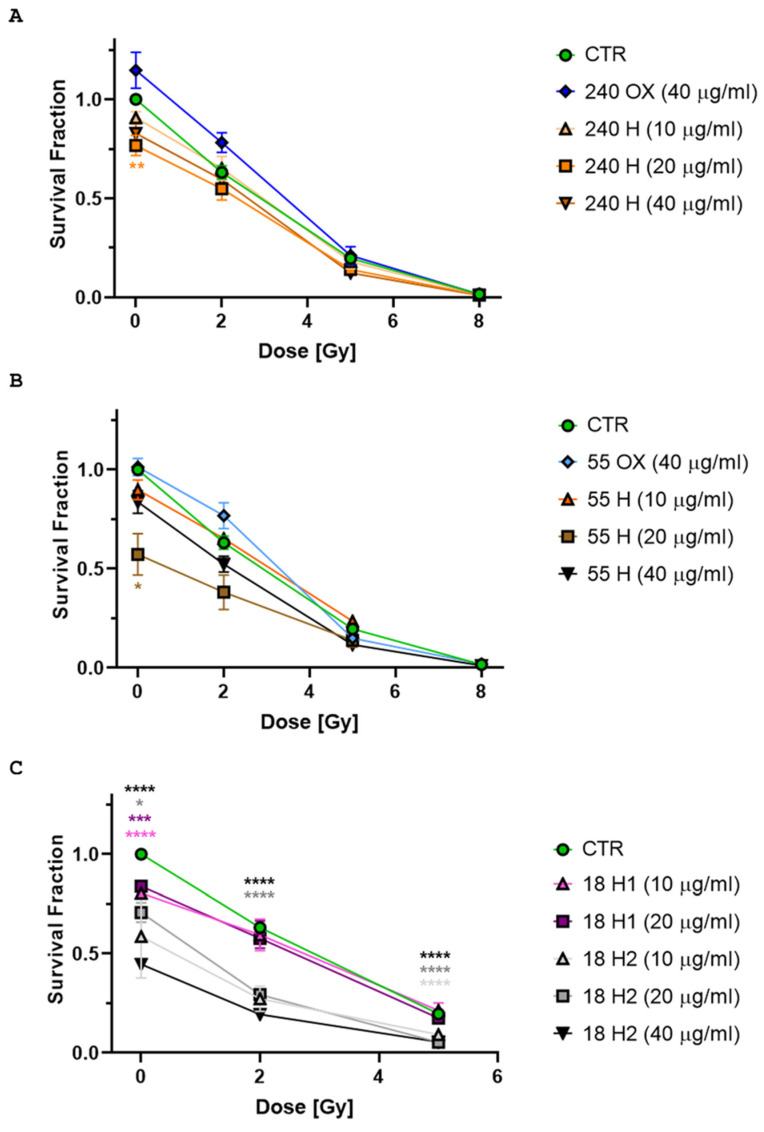
Clonogenic survival assays of DAOY cells exposed to NDs and X-rays (refer to Table 1 for ND sample specifications). (**A**) NDs of 240 nm, (**B**) NDs of 55 nm, and (**C**) NDs of 18 nm. Mean survival relative to parental untreated cells and standard error of the mean are shown. All experiments were performed on at least 10 replicates. Two-Way ANOVA tests with the Greenhouse–Geisser correction and Tukey’s multiple comparison tests were performed. The *p*-values ≤ 0.05 (*) and ≤0.01 (**) were considered as statistically significant; *p*-values ≤ 0.001 (***) and ≤0.0001 (****) were considered as highly statistically significant.

**Figure 6 ijms-24-16622-f006:**
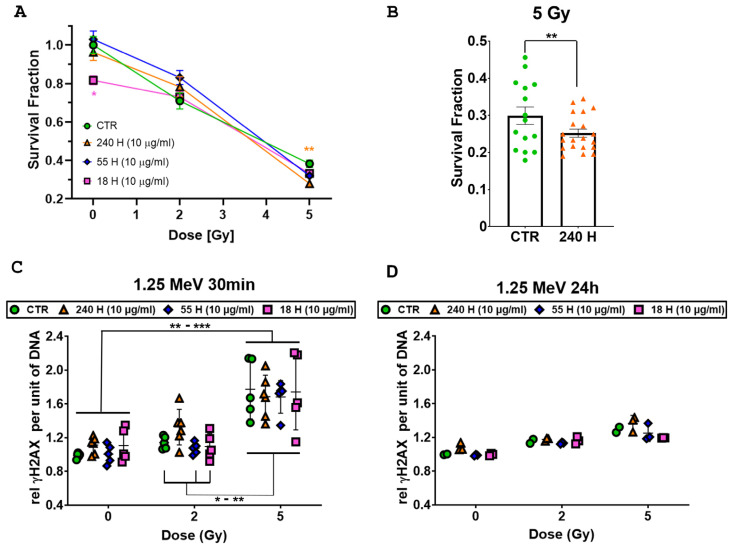
Effect of combined H-ND and 1.25 MeV γ-ray treatment on DAOY cells. (**A**) Clonogenic survival assays’ mean survival relative to parental untreated cells and standard error of the mean are shown. All experiments were performed on at least 10 replicates. (**B**) Histogram plot highlighting the differences between cells irradiated with 5 Gy pre-treated with 240 nm H-NDs (orange triangles) or untreated (CTR, green dots). (**C**) γ-H2AX flow cytometry assay at 30 min after irradiation. (**D**) γ-H2AX flow cytometry assay at 24 h after irradiation. Data were represented as the mean florescence intensity (MFI) per unit of DNA normalized on the sham-irradiated untreated controls. Two-way ANOVA tests with the Greenhouse–Geisser correction and Tukey’s multiple comparison tests have been performed. The *p*-values ≤ 0.05 (*) and ≤0.01 (**) were considered as statistically significant; the *p*-values ≤ 0.001 (***) was considered as highly statistically significant.

**Figure 7 ijms-24-16622-f007:**
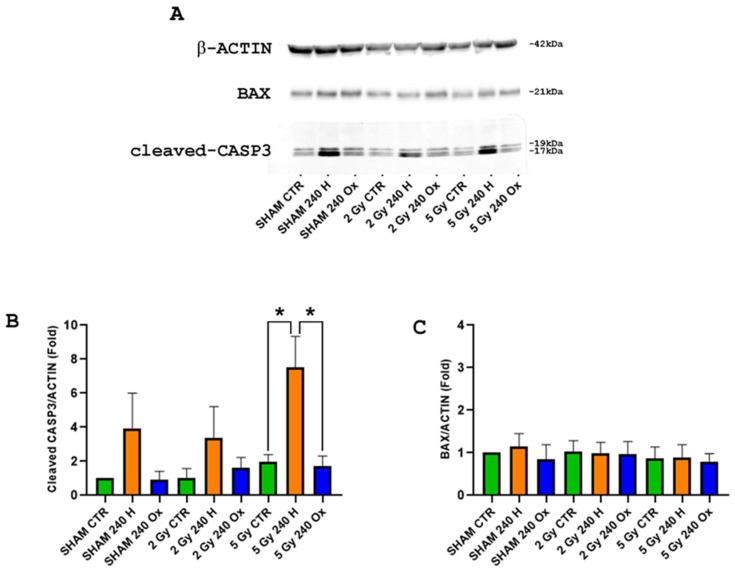
Western blot analyses showing representative images of Caspase-3, Bax and β-actin protein expression in DAOY cells from the sham, 240 nm H-NDs, and 240 nm OX-NDs exposed to 0, 2 and 5 Gy (*n* = 3) at 3 h post-irradiation. Band signals of target genes were normalized to those of β-actin. (**A**) The expression of the large cleavage form (17 kDa) of Caspase-3 increased significantly in DAOY cells treated with H-NDs and exposed to 5 Gy γ-rays. Bax was not modulated. (**B**,**C**) Quantification of Caspase-3 and Bax protein levels, showing fold changes vs. the sham controls. Student’s *t* test was performed. * *p* < 0.05.

**Figure 8 ijms-24-16622-f008:**
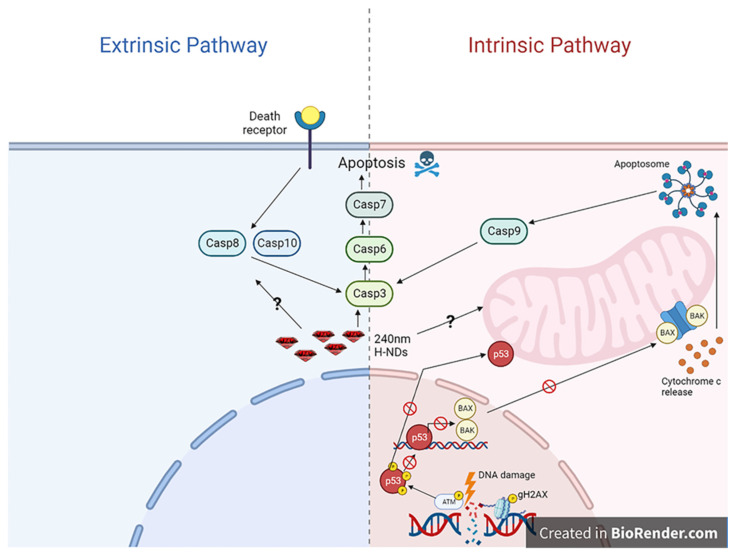
Representative schematic representation of extrinsic and intrinsic apoptosis pathways created with BioRender.com (accessed on 24 July 2023). H-NDs (red) of 240 nm diameter are localized in the perinuclear area. Possible interactions with molecular targets are shown.

**Table 1 ijms-24-16622-t001:** Summary of the names given to the ND samples, their median diameters and all the sequences of thermal treatments conducted on a given batch of NDs.

ND Sample Label	ND Median Diameter	Thermal Treatment Sequence
240 OX	240 nm	Annealing 2 h, 800 °C + Oxidation 12 h, 475 °C
240 H	240 nm	Annealing 2 h, 800 °C + Oxidation 12 h, 500 °C + Hydrogenation 3 h, 850 °C
55 OX	55 nm	Annealing 2 h, 800 °C + Oxidation 18 h, 500 °C
55 H	55 nm	Annealing 2 h, 800 °C + Oxidation 12 h, 500 °C + Hydrogenation 3 h, 750 °C
18 H = 18 H1	18 nm	Annealing 2 h, 800 °C + Oxidation 12 h, 500 °C + Hydrogenation 3 h, 750 °C
18 H2	18 nm	Annealing 2 h, 800 °C + Oxidation 12 h, 500 °C + Hydrogenation 6 h, 850 °C

**Table 2 ijms-24-16622-t002:** Specific forward and reverse primers for GAPDH and BAX.

Primers	
GAPDH forward	5′-ATTCCACCCATGGCAAATTC-3′
GAPDH reverse	5′-GGGATTTCCATTGATGACA-3′
BAX forward	5′-TGGCAGCTGACATGTTTTCTGAC-3′
BAX reverse	5′-TCACCCAACCACCCTGGTCTT-3′

## Data Availability

Other datasets analysed during the study are available from the corresponding authors on reasonable request.

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
