# Peer review of "Nanodiamond Effects on Cancer Cell Radiosensitivity: The Interplay between Their Chemical/Physical Characteristics and the Irradiation Energy"

_ijms, 2023, doi:10.3390/ijms242316622_

Round 1

Reviewer 1 Report

Comments and Suggestions for Authors

This paper presents evidence of the radiosensitizing effect of hydrogenated nanodiamonds. As the authors explicitly state, this attribute has not yet been extensively studied on this type of diamond, and only one other article on this matter exists in the literature (Grall et al.). Hence, this manuscript corroborates the previous observation of increased tumour cell mortality following simultaneous exposure to hydrogenated nanodiamonds and Gamma radiation. The current study distinguishes itself by implementing a distinct cell line, alternative nanodiamonds, and offering an alternative explanation for cell apoptosis. However, the radiosensitizing results are not as significant as those of Grall et al., and this manuscript does not provide novel perspectives on the source of this phenomenon. Nonetheless, the manuscript is well-written with detailed and comprehensive data. If the study may lack originality, it does provide value in confirming an unforeseen observation through an independent investigation and thus deserves publication, with consideration of the following remarks.

1- The parameters used by the authors to oxidize or hydrogenate their particles are those typically described in the literature. However, within the scope of this study, it is still necessary to provide characterizations of the modified particles to demonstrate that they are indeed hydrogenated oxidized (via FTIR and XPS for instance), and that the diamond core is still present (via Raman for instance). This is crucial to support the discussion on the effect of the particle's band structure on electron production.

2-  Flow cytometry is based on the fluorescence of diamond particles. It is necessary to provide a spectrum of this fluorescence."

3-What is the purpose of annealing at 800°C under an inert atmosphere?

4- Moreover, it is important to specify whether the annealing was conducted under N2, Ar, or vacuum, as these conditions are not equivalent. If it was performed to induce vacancy migration and promote the formation of NV centers, it would be beneficial to mention it. Similarly, the particles do not appear to have been irradiated to create vacancies; the authors are relying on those naturally present for N formation. It would be advisable to clarify this as well.

5-What were the characteristics of the suspensions of Ox-NDs and H-NDs? Did they aggregate in the same manner, and did they have the same sedimentation rate? These observations are important to consider when discussing the particle localization within cells. Additionally, hydrogenated nanodiamonds have a positive Zeta potential, while the oxidized ones have a negative potential. This is also a crucial factor to consider regarding the affinity of the particles for cellular membranes, for example.

6-The fluorescence properties can be influenced by the particle size and their synthesis method, and this is especially relevant when comparing the 18nm and 250nm particles (from two different suppliers). These factors should be taken into consideration in the discussion of fluorescence within the cells.

7-What bring the Raman mapping compared to fluorescence images ? It is unclear.

8-The higher toxicity of 18 nm H-NDs is suggested to be due to their tendency to aggregate. At present, without further characterizations beyond optical imaging, it is challenging to definitively conclude about this increased aggregation. Comparative DLS measurements would be necessary to provide evidence. Nevertheless, if toxicity is indeed related to the surface of H-NDs, it remains likely that size (i.e., the developed surface area) plays a significant role. The authors should take this point into consideration.

9-Similarly, without additional quantitative characterizations, one cannot conclude that there is more hydrogen on the nanodiamonds after annealing at 850°C compared to annealing at 750°C. The authors should take this point into consideration.

10-Figure 5a is intended to demonstrate the radiosensitizing effect of H-NDs. However, while I do not question the observed effect, it may not be very clear in this representation. The authors should consider providing an alternative visualization that better highlights the observed effect.

11-The authors mention in the discussion a stronger presence of graphitic phases on the small nanodiamonds compared to the large ones. This is not immediately evident, especially considering the treatments their particles have undergone (oxidation and hydrogenation). To clarify this point, the authors could rely on the Raman spectra of their particles (the G-band to diamond peak ratio).

12-Regarding the effect of water organization that might be responsible for the radiosensitizing phenomenon, the authors should refer to the physicochemical studies of the behavior of H-NDs under radiolysis that have already been conducted on the subject (E. Brun et al., Carbon N. Y. 162 (2020) 510–518. https://doi.org/10.1016/j.carbon.2020.02.063).

Reviewer 2 Report

Comments and Suggestions for Authors

In this research article, the authors have explored the use of commercially available nanodiamonds (NDs) and lab-modified types of NDs for radiosensitization applications. This area has been previously explored for radiosensitization applications in literature (doi: 10.1002/pssa.201700715; 10.3389/fonc.2023.1088878; 10.1016/j.biomaterials.2015.05.034). The authors have characterized the properties of NDs and explored biological applications in previous studies (doi: 10.3390/nano11102740; 10.1016/j.diamond.2018.11.001; 10.1002/advs.202301101). While the toxicity data is still limited to draw proper conclusions (doi: 10.1016/j.diamond.2009.11.022; 10.1016/j.envres.2018.05.027; 10.1021/nl1021909), nevertheless use of NDs in diagnostic and therapy applications have great potential due to their properties. The motivation is clear, but the structure of the manuscript can be bolstered. The authors have conducted relevant experiments and the data presented is within the scope of the journal, however the manuscript lacks basic physico-chemical characterization of nanoparticle materials. The article can be improved based on the comments below:

1.       Abstract does not attempt to explain use of oxidated NDs even though the manuscript states it. Abstract and manuscript need to be clearer on the topic (on the use of H-NDs and OX-NDs) and authors should make firm and convincing decision justifying the use one over the other. The manuscript can better explain the differences in chemical and structural properties between H and OX-NDs in terms of radiosensitization.

2.       Manuscript lacks an important figure schematic and workflow explaining the ND preparation and brief goals of the project. A clearly labelled synthesis and particle material schematic along with a work flowchart will be useful for readers.

3.       Please add multiple transmission electron microscopy (TEM) images at both low and high magnification for all particles discussed in this study in the supplementary file. Please add one selected image for each type in the manuscript. This data is very important for characterizing any nanoparticle. TEM size histograms can be very useful as well.

4.       Please add relevant dynamic light scattering (DLS) data values (curves can be added in supplementary) and polydispersity index (PdI) values (with relevant standard deviation values) to the manuscript. Zeta potentials with standard deviation values for various nanoparticles should be added.

5.       Have the authors performed any stability studies for the nanoparticles in relevant buffer solutions?

6.       The authors stated that they have explored different sizes for H and OX based NDs but only explored certain sizes of OX-NDs compared to H-NDs (as shown in Figures 1 and 2). Authors need to add relevant data for a proper comparison and analysis.

7.       Would recommend authors to add more images or observable evidence for the following statement:  “OX-NDs were primarily distributed throughout the cytoplasm, thanks to the presence of hydrophilic surface carboxylic groups, H-NDs, owing to their hydrophobic characteristics, adhered to the cell membranes.” to supplementary information.

8.       Figure 1C: Have the authors performed flow cytometry for samples of other sizes and between H vs OX particles. Such data should be added to the supplementary. The authors need to explain in the manuscript why 55 nm H-NDs were selected as the final candidate and why they were not compared with 55 nm OX-ND.

9.       Figure 1D: Similarly, the authors have selected the different size and only one-type of (H-ND) nanoparticle for analyzing uptake plateau. The data lacks consistency and the authors should add relevant comparison metrics from other sizes/ types in the graph or use same particle across all experiments and justify its usage.

10.   Have the authors conducted Raman mapping for 55 nm and 18 nm particles for both types? Please justify their exclusion from manuscript and add the relevant data to supplementary information.

11.   Figure 4: Why the authors have not compared OX-type NDs of various concentrations?

12.   I would recommend the authors to add images of stained radiated cells to supplementary if available.

13.   The authors should perform MTT assay for all relevant particles (at various concentrations) in at least two cell lines in vitro and add IC50 data and curves. ND toxicity is not completely proven and data in published literature varies, appropriate toxicity studies and serum analysis would be required for stating its non-toxicity in clinical use.

14.   Have the authors measured Bcl2 values and compared with BAX? If yes, such data can be useful in the manuscript.

15.   Would recommend the authors to perform a cytochrome-C assay  and apoptosis flow assay to corroborate their Western Blot results.

16.   The authors can add a lane at the bottom of the blots shown in supplementary file and add the intensity values measured.

Comments on the Quality of English Language

Minor corrections may be required.

Reviewer 3 Report

Comments and Suggestions for Authors

The authors report an interesting study on the radiosensitizing capacity of nanodiamonds with different chemical/physical characteristics towards cancer cells, also investigating the involved molecular mechanisms. The paper is well written and organized and the reported results have relevance in molecular medicine. I believe that this manuscript can be considered for publication in this Journal, in the present form.

The authors report a study on the radiosensitizing ability of nanodiamonds with different chemical/physical characteristics, towards cancer cells, also investigating the involved molecular mechanisms. The reported results have relevance in molecular medicine.

The author investigated in detail the molecular mechanisms  and the specific targets involved in the biological interactions of cancer cells with nanodiamonds. At the best of my knowledge, these issues have not yet been reported.
In my opinion, no improvements are required. The conclusion is consistent with the evidence and arguments presented and addresses the main issues posed. The references are appropriate.  

Author Response

We thank the reviewer for his/her appreciation of our study and his/her comments about writing style and text structure.

Round 2

Reviewer 1 Report

Comments and Suggestions for Authors

In response to my previous comments, the authors have provided answers that seem relevant to me and have modified the manuscript accordingly. I therefore recommend that the present manuscript be published as is.  

Reviewer 2 Report

Comments and Suggestions for Authors

While certain experiments were not attempted, the authors have satisfactorily answered all the questions raised during the review.